# Monocyte/Lymphocyte Ratio and MCHC as Predictors of Collateral Carotid Artery Disease—Preliminary Report

**DOI:** 10.3390/jpm11121266

**Published:** 2021-12-01

**Authors:** Tomasz Urbanowicz, Michał Michalak, Anna Olasińska-Wiśniewska, Michał Rodzki, Aleksandra Krasińska, Bartłomiej Perek, Zbigniew Krasiński, Marek Jemielity

**Affiliations:** 1Cardiac Surgery and Transplantology Department, Poznan University of Medical Sciences, 61-848 Poznan, Poland; anna.olasinska@poczta.onet.pl (A.O.-W.); michal.rodzki@skpp.edu.pl (M.R.); bperek@ump.edu.pl (B.P.); mjemielity@poczta.onet.pl (M.J.); 2Department of Computer Science and Statistics, Poznan University of Medical Sciences, 61-848 Poznan, Poland; michal@ump.edu.pl; 3Department of Ophtalmology, Poznan University of Medical Sciences, 61-848 Poznan, Poland; alex.krasinska@gmail.com; 4Department of Vascular, Endovascular Surgery, Poznan University of Medical Science, 61-848 Poznan, Poland; zbigniew.krasinski@skpp.edu.pl

**Keywords:** MLR 1, MCHC 2, carotid stenosis 3

## Abstract

Background: Carotid artery disease accounts for 30% of ischemic strokes in the general population. Numerous biomarkers have been investigated for predicting either the progression or the severity of the disease. The aim of this retrospective study was to compare hematologic indices among patients referred for surgical interventions due to severe carotid disease. Methods: In total, 135 patients (87 (64.4%) men and 48 (35.6%) women) with a mean age of 70 ± 8 years who underwent surgical carotid intervention were enrolled into the study. Results: A Mann–Whitney test for independent samples revealed significant differences in monocyte to lymphocyte ratio (MLR) and mean corpuscular hemoglobin concentration (MCHC) between patients with one and two (collateral) carotid diseases. The cut-off value for MLR was 0.3 (AUC = 0.654, *p* = 0.048, 70.0% sensitivity and 74.6% specificity) and for MHCH was 21.6. (AUC = 0.730, *p* < 0.001, 70.0% sensitivity and 77.2% specificity). A multivariable model of logistic regression revealed two significant parameters for collateral carotid stenosis disease including MLR > 0.3 (OR 6.19 with 95% CI 2.02–19.01, *p* = 0.001) and MCHC > 21.6 (OR 7.76, 95% CI 2.54–23.72, *p* < 0.001). Conclusions: MLR above 0.3 and MCHC above 21.6 have predictive values for colleterial carotid stenosis and may be used as easily accessible indicators for atherosclerosis severity.

## 1. Introduction

Carotid artery disease accounts for 30% of ischemic strokes in the general population [1]. Numerous biomarkers have been proposed for the prediction of disease occurrence, progression, complications, and short- and long-term prognosis [2,3]. 

Inflammation plays a key role in pathogenesis of cardiovascular diseases [4,5]. Not only immune cells, but also cytokines and other biomedical markers are involved in the mechanisms of atherosclerosis progression. However, in daily practice, the use of sophisticated methods of cytokines and immune cells analysis is impossible and unavailable. In turn, whole blood count enables an easy and valuable assessment of inflammatory response. Monocytes play a crucial role in innate immunity, while lymphocytes represent the adaptive system. Therefore, the analysis of both types of cells together is of unique value. Several studies showed a significance of monocytes and lymphocytes as markers of chronic inflammation in coronary and peripheral artery disease [6,7]. 

Monocyte to lymphocyte ratio (MLR) reflects inflammatory response in different stages of atherosclerosis [7]. 

Another marker, mean corpuscular hemoglobin concentration (MCHC), is a ratio between hemoglobin and hematocrit. This hematological index has been postulated to be linked with chronic and acute coronary syndromes [8]. Luke et al. [8] hypothesized that inflammation related to acute coronary syndrome may lead to higher oxidative stress which causes hemolysis followed by an increase in MCHC value. The correlation between MCHC and intima media thickness of carotid arteries was presented by Fornal et al. [9]. 

Carotid artery disease, as a part of the systemic atherosclerotic process, also has inflammatory pathogenesis. Therefore, the assessment of common hematological indices, which is valuable for patients with coronary artery disease, should also reflect carotid artery disorders. 

The aim of the study was to retrospectively compare hematological indices among patients referred for surgical interventions due to severe carotid disease.

## 2. Materials and Methods

In total, 135 patients (87 (64.4%) men and 48 (35.6%) women) with a mean age of 70 ± 8 years who were admitted to the Department of Vascular, Endovascular Surgery, Angiology and Phlebology in 2021 for carotid artery intervention, were enrolled into the study. There were 115 (85.2%) patients with single artery disease and 20 (14.8%) with collateral carotid stenosis (atheroslerotic changes diagnosed on both sides). Demographical and clinical data, including traditional risk factors for carotid disease, were collected and the severity of carotid stenosis was scrupulously evaluated by the extent of the disease and the percentage of artery lumen narrowing, as presented in Table 1. The atherosclerotic plaques narrowing above 30% of lumen of internal carotid artery (ICA) were regarded as significant for atherosclerosis involvement in ultrasound imaging. 

All patients were provided with a form of written informed consent for hospitalization and the research was conducted in accordance with the principles laid down in the Declaration of Helsinki. The patients were first enlisted into the study by verification of their survival, and their participation was voluntary, subject to being informed of the study. 

Carotid ultrasonography was carried out by qualified radiologists. Laboratory results were obtained on admission, and any clinical signs of infection of chronic inflammatory processes or oncologic history disqualified patients from the study. The whole blood count was the only standard laboratory test presenting inflammatory reaction. The other inflammatory parameters, such as C-reactive protein (CRP) or procalcitonin, were assessed only if infection was suspected. MLR was calculated as the ratio of monocyte to lymphocyte counts. Laboratory tests results are outlined in Table 2.

Statistical Analysis:

Continuous data were presented as means and standard deviations or medians and interquartile ranges [Q1–Q3] in case data did not follow the normal distribution (Shapiro–Wilk test). Nominal data were presented as numbers and percentages. The comparison between patients with one and two (collateral) carotid diseases was performed with the use of a Mann–Whitney test or a chi-square test for independence. The ROC analysis was performed in order to find an optimal cut-off point for continuous parameters. The parameter is considered to have prognostic properties if the area under the ROC curve (AUC) significantly differed from 0.5. The optimal cut-off point is determined by the Youden index (optimal cut-off point = max (sensitivity + specificity − 1)). A logistic regression both as univariable and multivariable analysis was performed to assess which parameters could be the predictors for the risk of collateral carotid disease. The results were presented as odds ratios (OR) and its 95% CI. The multivariable logistic regression model was denoted with the use of stepwise (backward selection) logistic regression analysis.

The analysis was performed with the use of the statistical package MedCalc^®^ Statistical Software version 19.6 (MedCalc Software Ltd., Ostend, Belgium; https://www.medcalc.org; 2020, access on date: 10 September 2021). All tests were considered significant at *p* < 0.05.

## 3. Results

A total of 135 patients were analyzed; 62 (46%) of them underwent ischemic events (56 (42%) stroke and 6 (4%) transient ischemic attack (TIA)). There were 23 (17%) patients suffering from vertigo, 2 (1.5%) from tinnitus, 2 (1.5%) from chronic headaches, and 1 (0.7%) from aphasia. 

There were 115 (85.2%) patients with single artery disease and 20 (14.8%) with collateral carotid stenosis. Neither cholesterol serum level including fractions nor fibrinogen serum levels nor kidney function tests differentiated the subgroups.

A Mann–Whitney test for independent samples revealed significant differences in MLR between patients with one and two (collateral) carotid diseases (*p* = 0.0288). The receiver operator curve (ROC curve) shows that MLR has prognostic properties for collateral carotid diseases AUC = 0.654, *p* value 0.048 with a 70% sensitivity and a 75% specificity, as presented in Figure 1. The optimal cut-off value was denoted as 0.3. Similarly, the Mann–Whitney test revealed significant differences in MCHC between patients with one and two (collateral) carotid diseases (*p* = 0.0116). The ROC analysis has shown that the optimal cut-off points for predicting collateral carotid disease is MHCH > 21.6 (AUC = 0.73, *p* = 0.001) giving a sensitivity of 70% and a specificity of 77.2%, as presented in Figure 2. 

Logistics regression analysis revealed that MLR values above 0.3 are significant predictors of advanced carotid disease involving both sides OR = 5.98, 95% CI 2.11–16.92. MCHC above 21.6 is also a significant predictor of collateral carotid disease OR = 7.52, 95% CI 2.63 –21.47, as presented in Table 3.

The multivariable model of logistic regression revealed two significant parameters for collateral carotid stenosis disease including MLR > 0.3 (OR = 6.20, 95% CI 2.02–19.01 and MCHC > 21.6 (OR = 7.76, 95% CI 2.54–23.72, as presented in Table 4.

## 4. Discussion

We present the results of our study that confirm the relation between MLR and the severity of carotid stenosis. Atherosclerosis is the process of chronic artery inflammation and the lymphocytes’ and monocytes’ role in all stages of atherosclerosis through inflammatory responses has been postulated [10]. Monocytes are a subset of leukocytes, which differentiate into macrophages when endothelial dysfunction occurs. They represent the strongest positive correlation with atherosclerosis [11]. 

Atherosclerosis as a chronic inflammatory disease is driven by immune response through defensive cells such as monocytes and macrophages. This autoimmune response against ApoB lipoproteins was detected in animal models with atherosclerosis [12]. With atherosclerosis progression, the protective force of an organism converts into a pathogenic one. It is clear that the early detection of patients with a tendency for excessive immunological response is essential. We believe that a simple marker such as MLR can be used as a useful predictor of advanced stenosis including both carotid arteries involvement. In our analysis, the ROC curves cut-off value of 0.30 for MLR predicted both carotid arteries involvement with a sensitivity of 68.18% and a specificity of 75% (ROC area under the curve: 0.658, 95% CI: 0.56–0.75, *p* = 0.0385). Previous reports postulated the relation between MLR and carotid severity in ischemic stokes but regarding one artery involvement [13]. 

Carotid artery stenosis occurrence due to chronic inflammatory process is the second largest cause of death globally [14]. To the best of our knowledge, no reports are available regarding the relationship between MLR and the collateral involvement of carotid atherosclerosis. Therefore, this study mainly focused on and demonstrated the relationship between MLR and both carotid artery stenosis. The utility of our finding is underlined by its informative value in the clinical assessment of patients referred for surgical intervention for carotid stenosis. Patients with higher MLR may potentially be at risk of repeated intervention due to the possible involvement of both carotid arteries, and, therefore, they should be under scrupulous follow-up after procedure. 

The clinical importance of MLR in cryptogenic ischemic stroke was postulated by Juega et al.’s analysis suggesting thromboembolic etiology [15]. Liu et al., in their retrospective analysis, revealed the predictive values of elevated MLR for strokes in patients with carotid disease [16]. Moreover, the high MLR was associated with worse clinical outcomes following a stroke episode by Ozgen et al. [17]. In our opinion, the higher MLR value may indicate patients with more advanced disease suggesting collateral involvement. Further studies evaluating the role of MLR in several medical strategies outcomes are needed.

Interestingly, MLR was also found to be useful in differentiating between patients hospitalized with fever due to bacterial infection and can help to undergo proper therapeutic steps in microbiology blood negative results [18]. We conclude that MLR can be a useful predictor for severity of carotid stenting, and during hospitalization can help to monitor patients with signs of infection. In Djordievic et al.’s study, the MLR was a very good independent predictor of lethal outcome in critically ill patients [19].

MLR is predominately reported as a marker of inflammation and infection and seems to be less related to endogenous stress [20]. Monocytes and macrophages infiltrating cells in myocarditis play a crucial role in the pathophysiology of the disease. Interestingly, in Mirna et al.’s study, the MLR correlated with the length of hospital stay with higher predictive characteristics than well-established biomarkers [21].

The result of our study presents mean corpuscular hemoglobin concentration (MCHC) as an another hematological index that was related to collateral carotid disease. This is the first study, to our best knowledge, presenting the relationship between MCHC and carotid disease. According to our study, multivariable analysis revealed MCHC (Odds ratio 2.8789 St. Err. 0.99742, 95% CI 1.4599–5.6774, *p* = 0.002) as a significant marker. We found the cut-off value of 21.3 for MCHC as a predictor of collateral carotid disease OR = 7.52, 95% CI 2.63–21.47.

A similar correlation was previously investigated in ischemic coronary disease. The higher values of MCHC were found during acute coronary syndromes [8]. In chronic coronary syndromes, the MCHC disturbances may be explained by the theory of complex interaction between inflammation, iron metabolism, and anemia, which affect MCHC value. Oxidative stress may also impair erythrocyte’s metabolism and result in hemolysis in acute coronary syndromes [22]. Fornal et al. presented the hypothesis of target organ damage in hypertension accompanied by erythropoiesis impairment, but Zhan et al. in their study found the relation only between red cells distribution and organ damage [9,23].

Limitation:

This is single center study, and all data was collected retrospectively. The analyzed laboratory parameters were obtained from the date of admission and represent the single center values.

## 5. Conclusions

MLR above 0.3 and MCHC above 21.3 have predictive values for collateral carotid stenosis and may be used as easily accessible indicators for atherosclerosis severity.

## Figures and Tables

**Figure 1 jpm-11-01266-f001:**
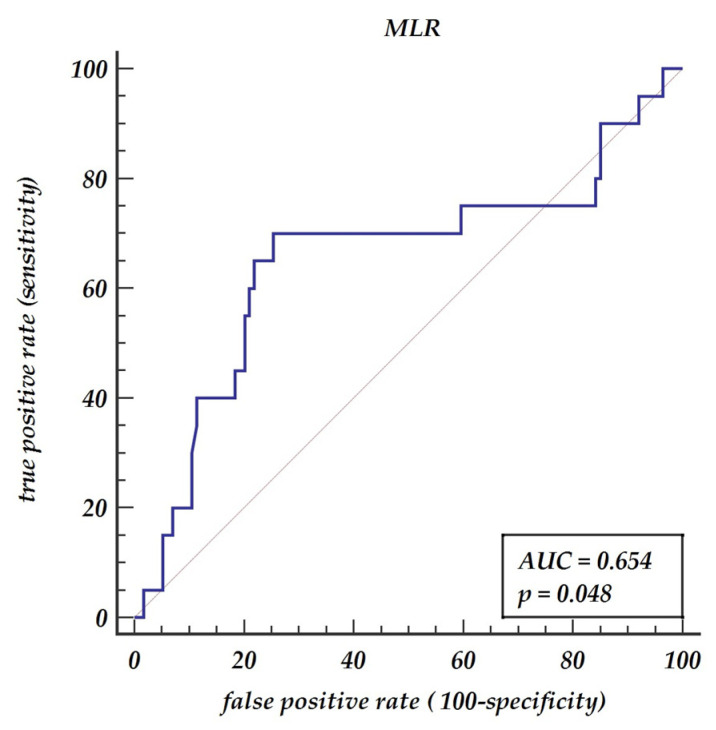
Receiver operator characteristic curve for monocyte to lymphocyte ratio. Abbreviations: AUC—area under the curve, MLR—monocyte to lymphocyte ratio.

**Figure 2 jpm-11-01266-f002:**
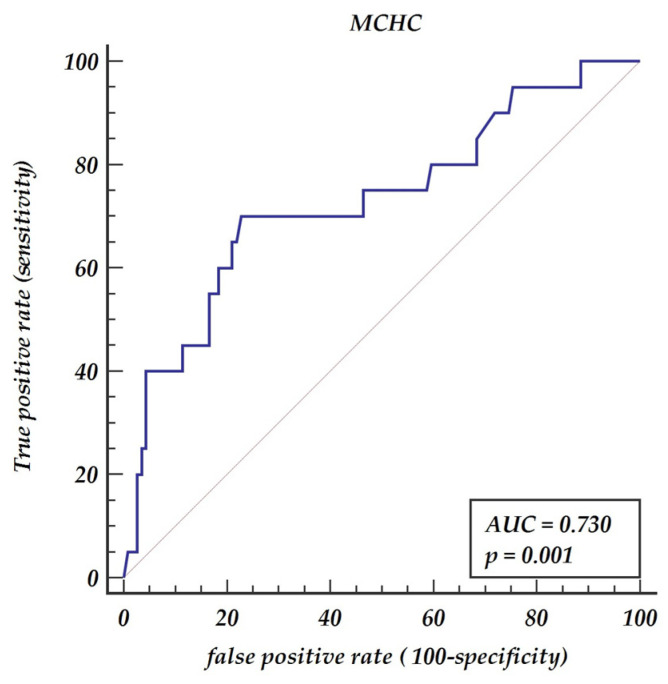
Receiver operator characteristic curve for mean corpuscular hemoglobin concentration (MCHC). Abbreviations: AUC—area under the curve, MCHC—mean corpuscular hemoglobin concentration.

**Table 1 jpm-11-01266-t001:** Demographical and clinical details.

	One Vessel Disease No = 115 (85.2%)	Collateral Disease No = 20 (14.8%)	*p*
Demographical:			
1. Age (median (Q1–Q3))	71 (66–75)	68 (64–72)	*p* = 0.1860
2. Gender (M(%)/F(%))	73(63.5%)/42(36.5%)	14 (70%)/6 (30%)	*p* = 0.5753
2. BMI (median (Q1–Q3))	26 (24.1–29.6)	26.82(24.5–29.3)	*p* = 0.8811
Carotid artery disease			
1. L(%)/R(%)	57 (49.6%)/58 (50.4%)		
2. Artery occlusion	10 (8.7%)	8 (40.0%)	*p* < 0.0001 *
3. Symptoms	70 (60.9%)	16 (80%)	*p* = 0.41
Concomitant disease			
1. Hypertension	90 (78.3%)	14 (70%)	*p* = 0.4176
2. Hypercholesterolemia	71 (65%)	23 (65%)	*p* = 0.9746
3. stroke	53 (46.1%)	9 (45%)	*p* = 0.7933
4. DM	39 (33.9%)	9 (45%)	*p* = 0.3391
5. CCS	37 (35%)	12 (33%)	*p* = 0.9849
6. COPD	6 (5.22%)	1 (5%)	*p* = 0.9677
7. Smoking	18 (19.8%)	2 (11.8%)	*p*= 0.4348
8.Atrial fibrillation	18 (15.7%)	3 (15.0%)	*p*= 0.9408

Abbreviations: BMI—body mass index, CCS—chronic coronary syndrome, COPD—chronic obstructive pulmonary disease, DM—diabetes mellitus, GFR—glomerular filtration rate, HDL—high-density lipoprotein, L—left, LDL—low-density lipoprotein, MPV—mean platelet volume, Plt—platelets, R—right, WBC—white blood count. *—statistically significant *p*-value.

**Table 2 jpm-11-01266-t002:** Laboratory results on admission.

	One Vessel Disease No = 104,115 (85.2%)	Collateral Disease No = 20 (14.8%)	*p*-Value
Whole blood count:			
1. WBC × 10^9^/L (median (Q1–Q3))	8.5 (6.7–9.8)	7.5 (6.4–11.3)	*p* = 0.8489
2. Neutrophils × 10^9^/L (median (Q1–Q3))	5.3 (4.1–6.6)	5.3 (4.2–7.6)	*p* = 0.5001
3. Lymphocyte × 10^9^/L (median (Q1–Q3))	2.0 (1.6–2.5)	1.7 (1.3–2.1)	*p* = 0.0367 *
4. Monocyte × 10^9^/L (median (Q1–Q3))	0.5 (0.4–0.7)	0.5 (0.4–0.7)	*p* = 0.8465
4. Eosinophils × 10^9^/L (median (Q1–Q3))	0.13 (0.08–0.18)	0.2 (0.07–0.21)	*p* = 0.8907
5. Basophils × 10^9^/L (median (Q1–Q3))	0.04 (0.03–0.06)	0.05 (0.03–0.07)	*p* = 0.5242
6. Luc × 10^9^/L (median (Q1–Q3))	0.16 (0.12–0.2)	0.15 (0.1–0.26)	*p* = 0.3425
7. Hemoglobin (mmol/L) (median (Q1–Q3))	8.7 (8.1–9.2)	9 (8.8–9.6)	*p* = 0.0139 *
5. Rbc × 10^9^/L (median (Q1–Q3))	4.45 (4.2–4.6)	4.56 (4.2–4.92)	*p* = 0.1381
6. Hematocrit (%) (median (Q1–Q3))	41.0 (39.0–43.0)	42.5 (40.0–43.0)	*p* = 0.3549
7. MCV (fL) (median (Q1–Q3))	92 (90–95)	91 (88–95)	*p* = 0.2084
8. MCH (pg) (median (Q1–Q3))	1.96 (1.89–2.02)	2 (1.94–2.05)	*p* = 0.1564
9. MCHC (mmol/dL) (median (Q1–Q3))	21 (20.7–21.5)	22 (21–22.6)	*p* = 0.0116 *
10. RDW (%) (median (Q1–Q3))	13.4 (13–14.1)	13.4 (12.9–13.8)	*p* = 0.5806
11. Plt × 10^3^/uL (median (Q1–Q3))	228 (182–292)	228 (190–267)	*p* = 0.4518
12. MPV (fL) (median (Q1–Q3))	7.9 (7.3–8.9)	8 (7.3–8.5)	*p* = 0.9776
13. MLR (median (Q1–Q3))	0.25 (0.21–031)	0.35 (0.26–0.51)	*p* = 0.0288 *
14. PDW (fL) (median (Q1–Q3))	58 (52–65)	59 (54–65)	*p* = 0.8489
Lipidogram:			
1. total cholesterol mmol/L (median (Q1–Q3))	3.9 (3.2–4.5)	4.1 (3.4–4.7)	*p* = 0.3571
2. HDL mmol/L (median (Q1–Q3))	1.1 (0.9–1.5)	1.2 (1–1.4)	*p* = 0.4098
3. LDL mmol/L (median (Q1–Q3))	2.2 (1.8–2.7)	2.4 (1.8–3.1)	*p* = 0.3619
Kidney function test:			
1. Creatinine mmol/L (median (Q1–Q3))	86 (69–106)	81 (69–116)	*p* = 0.7242
2. GFR (median (Q1–Q3))	76 (54–90)	80 (56–90)	*p* = 0.5208
Fibrinogen (mg/dL) (median (Q1–Q3))	364 (311–426)	355 (300–418)	*p* = 0.6257

Abbreviations: BMI—body mass index, COPD—chronic obstructive pulmonary disease, DM—diabetes mellitus, GFR—glomerular filtration rate, HDL—high-density lipoprotein, IHD—ischemic heart disease, L—left, Luc—large unstained cells, LDL—low-density lipoprotein, MPV—mean platelet volume, Plt—platelets, R—red blood cells, WBC—white blood cells. *—statistically significant *p*-value.

**Table 3 jpm-11-01266-t003:** Linear regression for collateral carotid disease.

	Odds	Std. Err.	z	*p* > z	95% Conf. Interval
Age	0.9713	0.0292	−0.97	0.334	0.9157–1.0303
BMI	0.9975	0.061	−0.04	0.967	0.8847–1.1246
Concomitant diseases:					
1. ischemic heart disease	1.0096	0.5129	0.02	0.985	0.3729–2.7328
2. Stroke	0.8799	0.4297	−0.26	0.793	0.3379–2.2913
3. Hypertension	0.6481	0.3485	−0.81	0.420	0.2258–1.859
4. DM	1.5944	0.7824	0.95	0.342	0.6094–4.1717
Whole blood count:					
1. WBC	1.1118	0.0967	1.22	0.223	0.9374–1.319
2. Neutrophils	1.1879	0.1178	1.74	0.082	0.9781–1.4427
3. Monocytes	2.7299	3.608	0.76	0.447	0.2047–36.4072
4. MLR	11.5519	15.9476	1.77	0.076	0.7719–172.8806
5. MLR > 0.3	5.9792	3.1726	3.37	0.001 *	2.1134–16.9158
6. Hb	2.1207	0.7078	2.25	0.024 *	1.1025–4.0792
7. MCHC	3.1288	1.01596	3.51	<0.001 *	1.6553–5.9122
8. MCHC > 21.6	7.52	4.02	3.77	<0.001 *	2.63–21.47
9. MCV	0.9512	0.0434	−1.1	0.273	0.8698–1.0403
10.Plt	0.9959	0.0035	−1.18	0.239	0.9891–1.0027
Serum cholesterol:					
1. Total	1.1743	0.2531	0.75	0.456	0.7698–1.7915
2. HDL	1.0955	0.7037	0.14	0.887	0.3111–3.858
3. LDL	1.2282	0.2752	0.92	0.359	0.7917–1.9053
GFR	1.0079	0.0129	0.61	0.541	0.9828–1.0336
Fibrinogen	0.9988	0.0025	−0.45	0.651	0.9941–1.0037

Abbreviations: BMI—body mass index, COPD—chronic obstructive pulmonary disease, DM—diabetes mellitus, GFR—glomerular filtration rate, HDL—high-density lipoprotein, IHD—ischemic heart disease, L—left, Luc—large unstained cells, LDL—low-density lipoprotein, MPV—mean platelet volume, Plt—platelets, R—red blood cells, WBC—white blood cells. *-statistically significant.

**Table 4 jpm-11-01266-t004:** Multivariable analysis.

	Odds	Std. Err.	z	*p* > z	95% Conf. Interval
MLR > 0.3	6.20	3.54	3.19	0.001	2.02–19.01
MCHC > 21.6	7.76	4.42	3.60	<0.001	2.54–23.72

Abbreviations: MCHC—mean corpuscular hemoglobin concentration, MLR—monocyte to lymphocyte ratio.

## Data Availability

All data will be available from the correspondence e-mail address for 3 years following the publication after reasonable request.

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
