# Peer review of "Monocyte/Lymphocyte Ratio and MCHC as Predictors of Collateral Carotid Artery Disease—Preliminary Report"

_jpm, 2021, doi:10.3390/jpm11121266_

Round 1
Reviewer 1 Report
The article titled "Monocyte/Lymphocyte ratio and MCHC as predictors of collateral carotid artery disease – preliminary report." by Urbanowicz et. al. is an interesting article showing that both the Monocyte/Lymphocyte ratio (MLR) and MCHC are reliable and accessible indicators for predicting colleterial carotid stenosis. These accessible hematologic markers can be used for accessing atherosclerosis severity.
I suggest making few changes and additions that might make the article more comprehensive and interesting:
- Please provide more information in the introduction, mostly regarding the MLR and MCHC importance as hematological markers in the literature and try to connect how that led to the hypothesis of this study.
- Please revise the introduction to make the flow more comprehensive, for my taste the paragraphs had weak connecting sentences.
- In statistical analysis, please provide information how the graphs were built and what are the Y X axis legends stand for (relevance). For instance, you can add instead of 100-specifity “false positive rate (100-specifity)”.
- Tables, I think adding a significance symbol like (*) may help easily identifying the statistically significant parameters instead or in addition to having the P value in bold.
Author Response
Dear Reviewer,
Thank you for your valuable comments. We corrected the manuscript according to them.
Reviewer: Please provide more information in the introduction, mostly regarding the MLR and MCHC importance as hematological markers in the literature and try to connect how that led to the hypothesis of this study.
- The Introduction section was corrected according to Reviewer’s comment.
Please revise the introduction to make the flow more comprehensive, for my taste the paragraphs had weak connecting sentences.
- The Introduction section was corrected according to Reviewer’s comment.
In statistical analysis, please provide information how the graphs were built and what are the Y X axis legends stand for (relevance). For instance, you can add instead of 100-specifity “false positive rate (100-specifity)
- This is a standard graph presenting the results for ROC curve analysis. No matter which statistical package someone use the description of the x axis and y axis is always the same. The ROC analysis presents the point (x, y) as (sensitivity , 1-specificity) or (sensitivity, 100% -specificity) for each possible cut-off points. The sensitivity = TPR (true positive rate)= True positives/ all positives or 1-FNR (false negative rate), and the specificity = TNR (true negative rate) = True negatives / all negatives or 1- TPR. If the area under the curve (AUC) is significantly differed from 0.5 , then the parameter got prognostic properties. The optimal cut-off point is determined by Youden index (optimal cut-off point = max (sensitivity + specificity -1).
- The graphs were corrected according to Reviewer’s comment as well as the corrections in statistical analysis section .
Tables, I think adding a significance symbol like (*) may help easily identifying the statistically significant parameters instead or in addition to having the P value in bold.
- The tables were corrected according to Reviewer’s comment.
Kind regards
Tomasz Urbanowicz

Reviewer 2 Report
The authors are presented an observational study addressing the association of high-sensitivity CRP vascular risk factors and cardiovascular diseases. The manuscript contains relevant information and is within the scope of this journal. There are however some relevant points to be addressed.
The authors are presenting an interesting study addressing the role of Monocyte to Lymphocyte ratio and mean corpuscular hemoglobin concentration as predictive factors for bilateral (?) carotid stenosis. The concept is valuable, but there are many important points to be clarified.
Major
Methods:
the reason for hospitalization should be clarified
Definition of collateral carotid stenosis
The criteria for variable inclusion in the logistic regression should be defined.
Detailed description of how carotid stenosis was evaluated and graded.
Minor
Abbreviations should be defined at first mention in the abstract and in text (MLR and MCHC b..)
The reason why stroke patients were hospitalized in a cardiology department should be presented
Author Response
Dear Reviewer
Thank you for your valuable comments. We corrected the manuscript according to them.
Major
Methods:
the reason for hospitalization should be clarified
- We corrected the methodology section. The patients were admitted for carotid artery procedure
Definition of collateral carotid stenosis
- We added the definition
The criteria for variable inclusion in the logistic regression should be defined.
- For univariate logistic regression model we analyzed all possible parameters, than we performed a stepwise logistic regression (backward selection) to find the set of significant parameter for predicting carotid disease. The corrections have been made in the section statistical analysis
Detailed description of how carotid stenosis was evaluated and graded.
- We added the information in the methodology section
Minor
Abbreviations should be defined at first mention in the abstract and in text (MLR and MCHC b..)
- We corrected and defined the abbreviations.
The reason why stroke patients were hospitalized in a cardiology department should be presented
- The patients were hospitalized for carotid artery procedure in the department of vascular surgery. The paper is a part of larger project of the collaborative surgical/cardiosurgical/cardiological team
Best regards
Tomasz Urbanowicz

Round 2
Reviewer 2 Report
The authors have responded to all my comments. I do not have nay further comments.